# PatchComplete: Learning Multi-Resolution Patch Priors for 3D Shape Completion on Unseen Categories

**Yuchen Rao**     **Yinyu Nie**     **Angela Dai**

Technical University of Munich

{yuchen.rao, yinyu.nie, angela.dai}@tum.de

## Abstract

While 3D shape representations enable powerful reasoning in many visual and perception applications, learning 3D shape priors tends to be constrained to the specific categories trained on, leading to an inefficient learning process, particularly for general applications with unseen categories. Thus, we propose *PatchComplete*, which learns effective shape priors based on multi-resolution local patches, which are often more general than full shapes (e.g., chairs and tables often both share legs) and thus enable geometric reasoning about unseen class categories. To learn these shared substructures, we learn multi-resolution patch priors across all train categories, which are then associated to input partial shape observations by attention across the patch priors, and finally decoded into a complete shape reconstruction. Such patch-based priors avoid overfitting to specific train categories and enable reconstruction on entirely unseen categories at test time. We demonstrate the effectiveness of our approach on synthetic ShapeNet data as well as challenging real-scanned objects from ScanNet, which include noise and clutter, improving over state of the art in novel-category shape completion by 19.3% in chamfer distance on ShapeNet, and 9.0% for ScanNet. [1]

## 1 Introduction

The prevalence of commodity RGB-D sensors (e.g., Intel RealSense, Microsoft Kinect, iPhone, etc.) has enabled significant progress in 3D reconstruction, achieving impressive tracking quality [25, 17, 27, 8, 40, 12] and even large-scale reconstructed datasets [10, 4]. Unfortunately, 3D scanned reconstructions remain limited in geometric quality due to clutter, noise, and incompleteness (e.g., as seen in the objects in Figure 1). Understanding complete object structures is fundamental towards constructing effective 3D representations, that can then be used to fuel many applications in robotic perception, mixed reality, content creation, and more.

Recently, significant progress has been made in shape reconstruction, across a variety of 3D representations, including voxels [9, 13, 34], points [15, 43], meshes [39, 11, 32], and implicit field representations [28, 21, 14, 18, 31, 3, 16, 33]. However, these methods tend to rely heavily on strong synthetic supervision, producing impressive reconstructions on similar train class categories but struggling to generalize to unseen categories. This leads to an expensive compute and data requirement in adapting to new objects in different scenarios, whose class categories may not necessarily have been seen during training and so must be re-trained or fine-tuned for.

In order to encourage more generalizable 3D feature learning to represent shape characteristics, we observe that while different class categories may have very different global structures, local geometric structures are often shared (e.g., a long, thin structure could represent a chair leg, a table leg, a lamp

---

[1]Source code available here.

36th Conference on Neural Information Processing Systems (NeurIPS 2022).

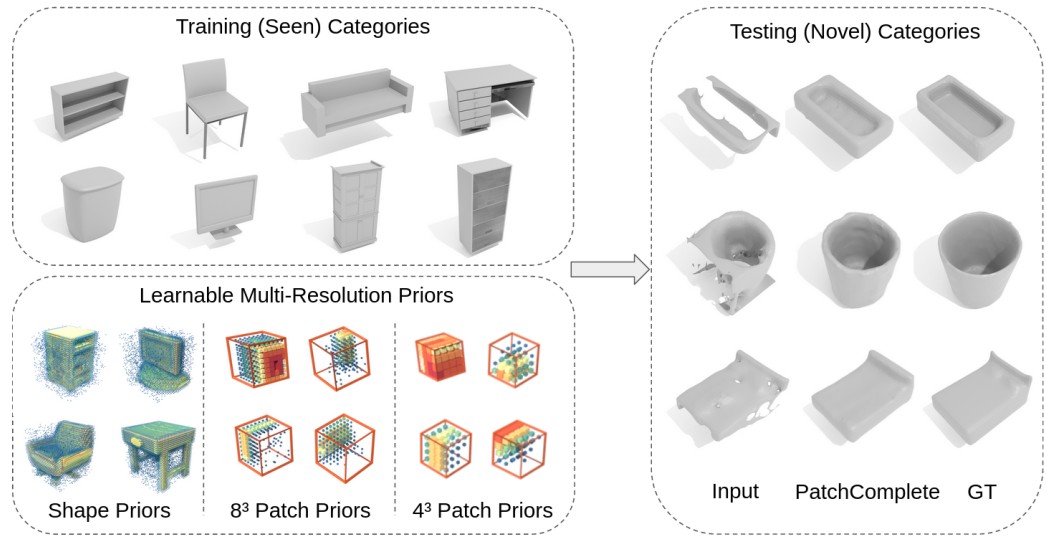

Figure 1: PatchComplete learns strong local priors for 3D shape completion by constructing multi-resolution patch priors from train shapes, which can then be applied for effective shape completion on unseen categories.

rod, etc.). We thus propose to learn a set of multi-resolution patch-based priors that captures such shared local substructures across the training set of shapes, which can be applied to shapes outside of the train set of categories. Our local patch-based priors can thus capture shared local structures, across different resolutions, that enable effective shape completion on novel class categories of not only synthetic data but also challenging real-world observations with noise and clutter.

We propose PatchComplete, which first learns patch priors for shape completion by correlating regions of observed partial inputs to the learned patch priors through an attention-based association, and decoding to reconstruct a complete shape. These patch priors are learned at different resolutions to encompass potentially different sizes of local substructures; we then learn to fuse the multi-resolution priors together to reconstruct the output complete shape. This enables learning generalizable local 3D priors that facilitate effective shape completion even for unseen categories, outperforming state of the art in synthetic and real-world observations by 19.3% and 9.0% on Chamfer Distance.

In summary, our contributions are:

- We propose generalizable 3D shape priors by learning patch-based priors that characterize shared local substructures that can be associated with input observations by cross-attention. This intermediate representation preserves structure explicitly, and can be effectively leveraged to compose complete shapes for entirely unseen categories.

- We design a multi-resolution fusion of different patch priors at various resolutions in order to effectively reconstruct a complete shape, enabling multi-resolution reasoning about the most informative learned patch priors to recover both global and local shape structures.

## 2   Related Work

### 2.1   3D Shape Reconstruction and Completion

Understanding how to reconstruct 3D shapes is an essential task for 3D machine perception. In particular, the task of shape completion to predict a complete shape from partial input observations has been studied by various works toward understanding 3D shape structures. Recently, many works have leveraged deep learning techniques to learn strong data-driven priors for shape completion, focusing on various different representations, e.g. volumetric grids [41, 13], continuous implicit functions [29, 21, 28, 7, 36], point clouds [30, 33], and meshes [11, 19]. These works tend to focus on strong synthetic supervision on a small set of train categories, achieving impressive performance on unseen shapes from train categories, but often struggling to generalize to unseen classes. We focus

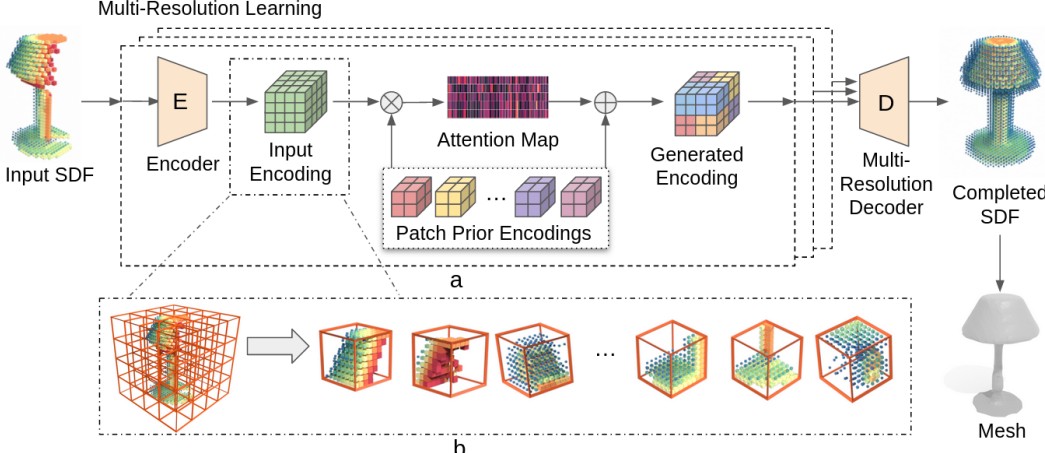

Figure 2: Overview of our approach. (a) shows our multi-resolution prior learning for shape completion. Each dotted block indicates patch prior learning for a single resolution, and the three different resolution encodings are then fused in a multi-resolution decoder that outputs a complete shape as an SDF grid. (b) illustrates our input partial observations and local input patches, from which we learn mappings to learned patch priors as to how to best compose the complete shape.

on learning more generalizable, local shape priors in order to effectively reconstruct complete shapes on unseen class categories.

## 2.2 Few-Shot and Zero-Shot 3D Shape Reconstruction

Several works have been developed to tackle the challenging task of few-shot or zero-shot shape reconstruction, as observations in-the-wild often contain a wide diversity of objects. In the few-shot scenario where several examples of novel categories are available, Wallace and Hariharan [38] learn to refine a given category prior. Michalkiewicz et al. [22] further propose to learn compositional shape priors for single-view reconstruction. In the zero-shot scenario without any examples seen for novel categories, Naeem et al. [24] learn priors from seen categories to generate segmentation masks for unseen categories. Zhang et al. [44] additionally proposed to use spherical map representations to learn priors for the reconstruction of novel categories. Thai et al. [35] recently developed an approach to transfer knowledge from an RGB image for shape reconstruction. We also tackle a zero-shot shape reconstruction task, by learning a multi-resolution set of strong local shape priors to compose a reconstructed shape.

Several recent works have explored learning shape priors by leveraging a VQ-VAE backbone with autoregressive prediction to perform shape reconstruction [23, 42]. In contrast, we propose to learn multi-resolution shape priors without requiring any sequence interpretation, which enables direct applicability to real-world scan data that often contains noise and clutter. Additionally, hierarchical reconstruction has shown promising results for shape reconstruction [2, 6]. Our approach also takes a multi-resolution perspective, but learns explicit shape and local patch priors and their correlation to input partial observations for robust shape completion.

## 3 Method

### 3.1 Overview

Our method aims to learn effective 3D shape priors that enable general shape completion from partial input scan data, on novel class categories not seen during training. Key to our approach is the observation that 3D shapes often share repeated local patterns – for instance, chairs, tables, and nightstands all share a support surface, and chair or table legs can share a similar structure with lamp rods. Inspired by this, we regard a complete object as a set of substructures, where each substructure represents a local geometric region. We thus propose PatchComplete to learn such local priors and assemble them into a complete shape from a partial scan. An overview of our approach is shown in Figure 2.

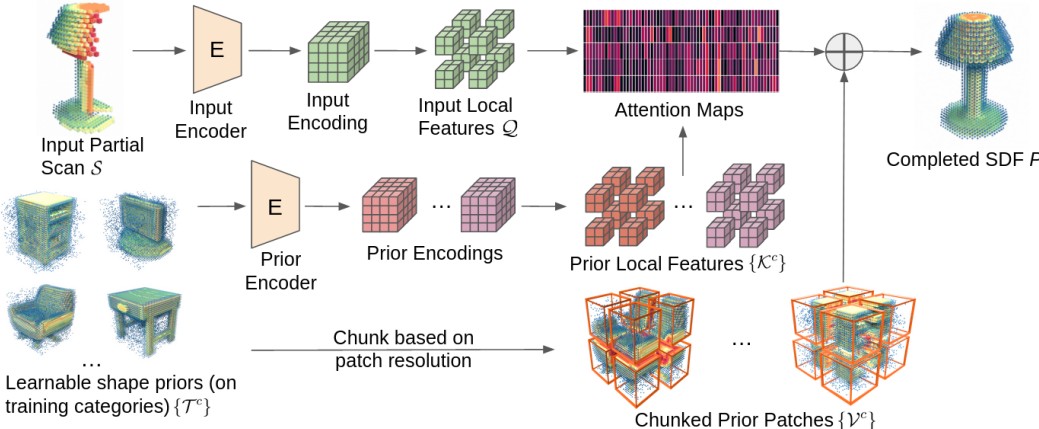

Figure 3: Network architecture for local patch-based shape prior learning under a single resolution. We learn to build mappings from local regions in a partial object scan $\mathcal{S}$ to local priors based on complete learnable shape priors $\{\mathcal{T}^c\}$. An input encoder takes the input of $\mathcal{S}$ to encode its local features (represented by patches $\mathcal{Q}$). Analogously, we adopt a prior encoder to process each prior in $\{\mathcal{T}^c\}$ to construct a local prior feature pool $\{\mathcal{K}^c\}$. In parallel, we chunk priors in $\{\mathcal{T}^c\}$ into patch volumes $\{\mathcal{V}^c\}$, which are correspondingly fused by the attention to input patches to compose the most informative patch priors: we use each incomplete local patch in $\mathcal{Q}$ to query the keys $\{\mathcal{K}^c\}$ from complete prior patches, and assemble their corresponding patch volumes $\{\mathcal{V}^c\}$ for shape completion.

## 3.2 Learning Local Patch-Based Shape Priors

We first learn local shape priors from ground-truth train objects. We represent both the input partial scan $\mathcal{S}$ and the ground-truth shape $\mathcal{G}$ as 3D truncated signed distance field grids of size $D^3$. Figure 3 illustrates the process of learning local shape priors. We learn to build mappings from local regions in incomplete input observations to local priors based on complete shapes, in order to robustly reconstruct the complete shape output.

We first aim to learn patch-based priors, which we extract from learnable global shape priors $\{\mathcal{T}^c\}$. We denote $\mathcal{G}^c$ as the set of ground-truth train objects in the $c$-th category. These $\{\mathcal{T}^c\}$ priors are initialized per train category $c$ based on mean-shift clustering within shapes in $\mathcal{G}^c$. Thus $\mathcal{T}^c$ is a set of representative samples in each category, which are encoded parallel to the input scan $\mathcal{S}$.

Both encoders are analogously structured as 3D convolutional encoders which spatially downsample by a factor of $R$, resulting in a 3D encoding of size $N^3$; $N = D/R$. The input encoding (from $\mathcal{S}$) are uniformly chunked into patches $\{\mathcal{Q}_i\}$, $i=1,...,N^3$; similarly, the encoded priors (from $\{\mathcal{T}^c\}$) are chunked into patches $\{\mathcal{K}_i^c\}$.

We then use each local part $\mathcal{Q}_i$ to query for the most informative encodings of complete local regions $\{\mathcal{K}_i^c\}$, building this mapping based on cross-attention [37]. Then for each input patch, we calculate its similarity with all local patches in representative shapes on training categories:

$$\text{Attention}(\mathcal{Q}_i, \mathcal{K}) = \text{softmax}(\frac{\mathcal{Q}_i \cdot \mathcal{K}^T}{d/2}), \mathcal{K} = \{\mathcal{K}_i^c | i = 1, ..., N^3, c = 1, ..., C\}, \quad (1)$$

where $d$ is the dimension of the encoded vectors of $\mathcal{Q}_i, \mathcal{K}_i^c$; $C$ is the category number. We then reconstruct complete shape patches $\{\mathcal{P}_i\}$ by:

$$\mathcal{P}_i = \text{Attention}(\mathcal{Q}_i, \mathcal{K}) \cdot \mathcal{V} \quad (2)$$

where $\mathcal{V} = \{\mathcal{V}_i^c\}$ is the set of $N^3$ chunks (of resolution $R$) from shapes in all categories $\{\mathcal{T}^c\}$, where each $\mathcal{V}_i^c$ is paired with $\mathcal{K}_i^c \in \mathcal{K}$. We can then recompose $\{\mathcal{P}_i\}$ to the predicted full shape $P$.

**Loss.** We use an $\ell_1$ reconstruction loss to train the learned local patch priors. Note that $\{\mathcal{T}^c\}$ are learned along with the network weights, enabling learning the most effective global shape priors for the shape completion task.

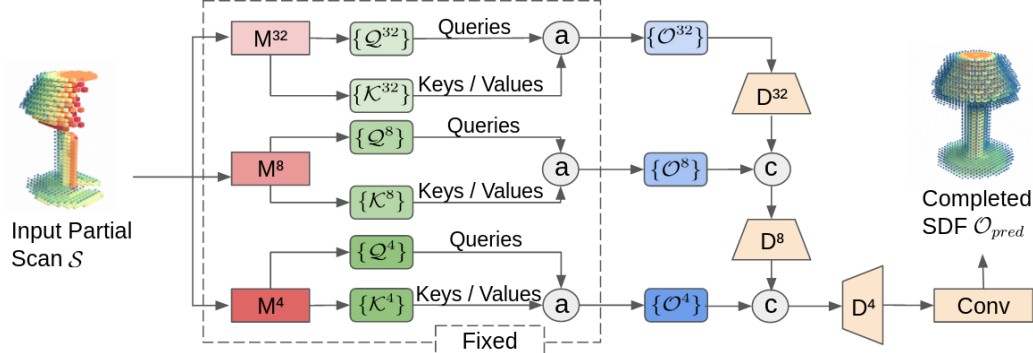

Figure 4: Network architecture for multi-resolution learning pipeline. We generate a complete shape from a partial input scan $\mathcal{S}$ by fusing input local features and learned local priors in a multi-scale fashion. We first extract the input local features $\{\mathcal{Q}^R\}$ and the learned local priors $\{\mathcal{K}^R\}$ using the prior learning model ($M^R$) in Section 3.2 under different resolutions ($R$=32, 8, 4). Then, we use attention (Equation 3) to generate intermediate features $\{\mathcal{O}^R\}$, which we then recursively fuse to decode a complete shape.

### 3.3 Multi-Resolution Patch Learning

In Section 3.2, we learn to complete shapes in a single patch resolution $R$. Since local substructures may exist at various scales, we learn patch priors under varying resolutions, which we then use to decode a complete shape. We use three patch resolutions ($R$=4, 8, 32), for which we learn patch priors. This results in three pairs of trained {input encoder, prior encoder} (see Figure 3). Given a partial input scan $\mathcal{S}$, each pair outputs a set of 1) input patch features $\{\mathcal{Q}_i^R\}$, and 2) prior patch features $\mathcal{K}^R$, under the resolution of $R$=4, 8, 32. In this section, we decode a complete shape from these multi-resolution patch priors.

Since $\mathcal{K}^R$ stores the patch priors, we use it as the (key, value) term into an attention module, where each input patch feature $\mathcal{Q}_i^R$ is used to query for the most informative prior features in $\mathcal{K}^R$ under different resolutions, from which we complete partial patches in feature spaces with a multi-scale fashion. We formulate this process by

$$\mathcal{O}_i^R = \text{Concat}\left[\mathcal{Q}_i^R, \text{Attention}(\mathcal{Q}_i^R, \mathcal{K}^R) \cdot \mathcal{K}^R\right], R = 4, 8, 32, \tag{3}$$

where we concatenate input patch feature with the attention result to compensate information loss on observable regions. This outputs $\{\mathcal{O}_i^R\}$ as the intermediate feature of the $i$-th patch; $i = 1, ..., N_R^3$. $N_R$ denotes the number of patches under the resolution $R$; $N_R = D/R$. We recompose all generated patch features into a volume $\mathcal{O}^R = \{\mathcal{O}_i^R | i = 1, ..., N_R^3\}$. Note that the dimension of each grid feature in $\{\mathcal{Q}_i^R\}$ and $\mathcal{K}^R$ equals to $d$ (see Eq. 1). Then $\mathcal{O}^R$ is with the dimension of $N_R^3 \times 2d$.

We then sequentially use 3D convolution decoders to upsample and concatenate $\mathcal{O}^R$ from low to high resolution to fuse all the shape features as in

$$\mathcal{O}^{R_{r+1}} = \text{Concat}\left[\text{Upsample}(\mathcal{O}^{R_r}), \mathcal{O}^{R_{r+1}}\right], R_r|_{r=1,2,3} = 32, 8, 4,$$
$$\mathcal{O}_{pred} = \text{Conv}(\text{Upsample}(\mathcal{O}^{R_3})). \tag{4}$$

In Eq. 4, $\mathcal{O}^{R_r}$ has a lower resolution than $\mathcal{O}^{R_{r+1}}$. We then use deconvolution layers to upsample $\mathcal{O}^{R_r}$ to match the resolution of $\mathcal{O}^{R_{r+1}}$, and then concatenate them together. We recursively fuse $\mathcal{O}^{R_r}$ into $\mathcal{O}^{R_{r+1}}$, which produces $\mathcal{O}^{R_3}$ as the final fusion result with the resolution of $(D/4)^3$. An extra upsampling followed by a convolution layer is adopted to upsample $\mathcal{O}^{R_3}$ into our shape prediction $\mathcal{O}_{pred}$ with the dimension of $D^3$.

In training the feature fusion, we fix all parameters from the {input encoder, prior encoder} under the three resolutions, since they are pre-trained on Section 3.2 and provide better-learned priors and attention maps under these strong constraints. The whole pipeline for this section can be found in Figure 4.

We use an $\ell_1$ loss to supervise the TSDF value regression. We weight the $\ell_1$ loss to penalize false predictions based on the predicted signs as Eq. 5 shows, which represents whether this voxel grid is occupied or not. It is also used in the loss function in Section 3.2.

$$\mathcal{L} = w_{occ}\ell_1(\mathcal{O}_{pred}^{occ}, \mathcal{O}_{gt}) + w_{empty}\ell_1(\mathcal{O}_{pred}^{empty}, \mathcal{O}_{gt}) + \ell_1(\mathcal{O}_{pred}^{correct}, \mathcal{O}_{gt}). \tag{5}$$

In Eq. 5, $\mathcal{O}_{pred}^{empty}$ represents the false positive TSDF values, where the ground truth has negative signs and the prediction has positive signs, which in general indicates the missing predictions. $\mathcal{O}_{pred}^{occ}$ represents false negative TSDF values, where the ground truth has positive signs and the prediction has negative signs, which indicates extra predictions. $\mathcal{O}_{pred}^{correct}$ means those predicted TSDF values with the same signs as the ground-truth. During training, we choose the weight for false positive ($w_{empty}$) as 5, and the weight for false negative ($w_{occ}$) as 3.

### 3.4  Implementation Details

We train our approach on a single NVIDIA A6000 GPU, using an Adam optimizer with batch size 32 for a synthetic dataset and batch size 16 for a real-world dataset, and an initial learning rate of 0.001. We train for 80 epochs until convergence, and then decrease the learning rate by half after epoch 50. We use the same settings for learning priors and the multi-resolution decoding, which train for 4 and 15.5 hours respectively. For additional network architecture details, we refer to the supplemental material.

Note that for training for real scan data, we first pre-train on synthetic data and then fine-tune only the input encoder. We use 112 shape priors in our method, which are clustered from 3202 train shapes, and represented as TSDFs with truncation of 2.5 voxels.

## 4  Experiments and Analysis

### 4.1  Experimental Setup

**Datasets.**    We train and evaluate our approach on synthetic shape data from ShapeNet [5] as well as on challenging real-world scan data from ScanNet [10]. For ShapeNet data, we virtually scan the objects to create partial input scans, following [26, 13]. We use 18 categories during training, and test on 8 novel categories, resulting in 3,202/1,325 train/test models with 4 partial scans for each model.

For real data, we use real-scanned objects from ScanNet extracted by their bounding boxes, with corresponding complete target data given by Scan2CAD [1]. We use 8 categories for training, comprising 7,537 train samples, and test on 6 novel categories of 1,191 test samples.

For all experiments, objects are represented as $32^3$ signed distance grids with truncation of 2.5 voxel units for ShapeNet and 3 voxel units for ScanNet. The objects in ShapeNet are normalized into the unit cube, while we keep the scaling for ScanNet objects, and save their voxel sizes separately to keep the real size information. Additionally, to train and evaluate on real data, all methods are first pre-trained on ShapeNet and then fine-tuned on ScanNet.

**Baselines.**    We evaluate our approach against various state-of-the-art shape completion methods. We compare with state-of-the-art shape completion methods 3D-EPN [13] and IF-Nets [7], which learn effective shape completion on dense voxel grids and with implicit neural field representations, respectively, without any focus on unseen class categories. We further compare to the state-of-the-art few-shot shape reconstruction approach of Wallace and Hariharan [38] (referred to as Few-Shot) leveraging global shape priors, which we apply in our zero-shot unseen category scenario. Finally, AutoSDF [23] uses a VQ-VAE module with a transformer-based autoregressive model over latent patch priors to produce TSDF shape reconstruction.

**Evaluation Metrics.**    To evaluate the quality of reconstructed shape geometry, we use $\mathcal{L}_1$ Chamfer Distance (CD) and Intersection over Union (IoU) between predicted and ground truth shapes. To evaluate methods that output occupancy grids, we use occupancy thresholds used by the respective methods to obtain voxel predictions, i.e. 0.4 for [38] and 0.5 for [7]. To evaluate methods that output signed distance fields, we extract the iso-surface at level zero with marching cubes [20]. 10K points are sampled on surfaces for CD calculation. Both Chamfer distance and IoU are evaluated on objects in the canonical system, and we report Chamfer Distance scaled by $\times 10^2$.

Table 1: Quantitative comparison for shape completion on synthetic ShapeNet [5] data.

| | Chamfer Distance ($\times 10^2$)↓ | | | | | IoU↑ | | | | |
|---|---|---|---|---|---|---|---|---|---|---|
| | 3D-EPN [13] | Few-Shot [38] | IF-Nets [7] | Auto-SDF [23] | Ours | 3D-EPN [13] | Few-Shot [38] | IF-Nets [7] | Auto-SDF [23] | Ours |
| Bag | 5.01 | 8.00 | 4.77 | 5.81 | **3.94** | 0.738 | 0.561 | 0.698 | 0.563 | **0.776** |
| Lamp | 8.07 | 15.10 | 5.70 | 6.57 | **4.68** | 0.472 | 0.254 | 0.508 | 0.391 | **0.564** |
| Bathtub | 4.21 | 7.05 | 4.72 | 5.17 | **3.78** | 0.579 | 0.457 | 0.550 | 0.410 | **0.663** |
| Bed | 5.84 | 10.03 | 5.34 | 6.01 | **4.49** | 0.584 | 0.396 | 0.607 | 0.446 | **0.668** |
| Basket | 7.90 | 8.72 | **4.44** | 6.70 | 5.15 | 0.540 | 0.406 | 0.502 | 0.398 | **0.610** |
| Printer | 5.15 | 9.26 | 5.83 | 7.52 | **4.63** | 0.736 | 0.567 | 0.705 | 0.499 | **0.776** |
| Laptop | 3.90 | 10.35 | 6.47 | 4.81 | **3.77** | 0.620 | 0.313 | 0.583 | 0.511 | **0.638** |
| Bench | 4.54 | 8.11 | 5.03 | 4.31 | **3.70** | 0.483 | 0.272 | 0.497 | 0.395 | **0.539** |
| Inst-Avg | 5.48 $\pm 2e^{-1}$ | 9.75 $\pm 9e^{-2}$ | 5.37 $\pm 1e^{-1}$ | 5.76 $\pm 3e^{-2}$ | **4.23** $\pm 4e^{-2}$ | 0.582 $\pm 9e^{-3}$ | 0.386 $\pm 1e^{-3}$ | 0.574 $\pm 4e^{-5}$ | 0.446 $\pm 6e^{-3}$ | **0.644** $\pm 1e^{-3}$ |
| Cat-Avg | 5.58 $\pm 2e^{-1}$ | 9.58 $\pm 1e^{-1}$ | 5.29 $\pm 1e^{-1}$ | 5.86 $\pm 5e^{-3}$ | **4.27** $\pm 5e^{-2}$ | 0.594 $\pm 8e^{-3}$ | 0.403 $\pm 1e^{-3}$ | 0.581 $\pm 3e^{-4}$ | 0.452 $\pm 7e^{-3}$ | **0.654** $\pm 1e^{-3}$ |

## 4.2 Evaluation on Synthetic Data

In Table 1, we evaluate our approach in comparison with prior arts on unseen class categories of synthetic ShapeNet [5] data. Our approach on learning attention-based correlation to learned local shape priors results in notably improved reconstruction performance, with coherent global and local structures, as shown in Figure 5. In Table 1, our work outperforms other baselines both instance-wise and category-wise. One of the key factors is that our method learns multi-scale patch information from seen categories to complete unseen categories with enough flexibility, while most of the other baselines are designed for 3D shape completion on known categories, which hardly leverage shape priors across categories.

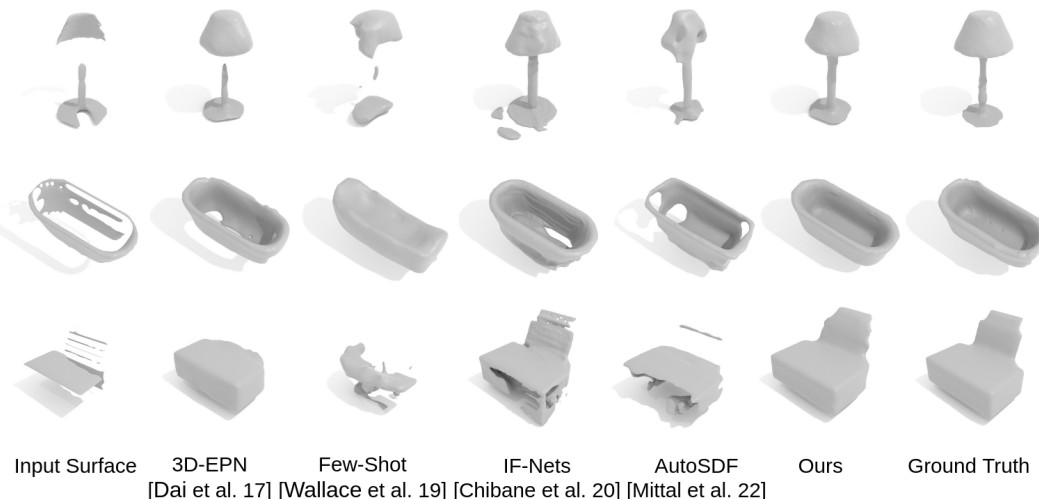

| Input Surface | 3D-EPN [Dai et al. 17] | Few-Shot [Wallace et al. 19] | IF-Nets [Chibane et al. 20] | AutoSDF [Mittal et al. 22] | Ours | Ground Truth |

Figure 5: Qualitative comparison for shape completion on synthetic ShapeNet [5] dataset.

## 4.3 Evaluation on Real Scan Data

Table 2 evaluates our approach in comparison with prior arts on real scanned objects from unseen categories in ScanNet [10]. Here, input scans are not only partial but often contain noise and clutter; our multi-resolution learned priors enable more robust shape completion in this challenging scenario. Results in Figure 6 further demonstrate that our approach presents more coherent shape completion than prior methods by using cross-attention with learnable priors, which better preserves the global structures in coarse and cluttered environments.

Table 2: Quantitative comparison with state of the art on real-world ScanNet [10] shape completion.

| | Chamfer Distance ($\times 10^2$)↓ | | | | | IoU↑ | | | | |
|---|---|---|---|---|---|---|---|---|---|---|
| | 3D-EPN [13] | Few-Shot [38] | IF-Nets [7] | Auto-SDF [23] | Ours | 3D-EPN [13] | Few-Shot [38] | IF-Nets [7] | Auto-SDF [23] | Ours |
| Bag | 8.83 | 9.10 | 8.96 | 9.30 | **8.23** | 0.537 | 0.449 | 0.442 | 0.487 | **0.583** |
| Lamp | 14.27 | 11.88 | 10.16 | 11.17 | **9.42** | 0.207 | 0.196 | 0.249 | 0.244 | **0.284** |
| Bathtub | 7.56 | 7.77 | 7.19 | 7.84 | **6.77** | 0.410 | 0.382 | 0.395 | 0.366 | **0.480** |
| Bed | 7.76 | 9.07 | 8.24 | 7.91 | **7.24** | 0.478 | 0.349 | 0.449 | 0.380 | **0.484** |
| Basket | 7.74 | 8.02 | 6.74 | 7.54 | **6.60** | 0.365 | 0.343 | 0.427 | 0.361 | **0.455** |
| Printer | 8.36 | 8.30 | 8.28 | 9.66 | **6.84** | 0.630 | 0.622 | 0.607 | 0.499 | **0.705** |
| Inst-Avg | 8.60 $\pm 2e^{-1}$ | 8.83 $\pm 2e^{-2}$ | 8.12 $\pm 7e^{-2}$ | 8.56 $\pm 2e^{-2}$ | **7.38** $\pm 6e^{-2}$ | 0.441 $\pm 2e^{-3}$ | 0.387 $\pm 1e^{-3}$ | 0.426 $\pm 3e^{-3}$ | 0.386 $\pm 1e^{-4}$ | **0.498** $\pm 9e^{-3}$ |
| Cat-Avg | 9.09 $\pm 3e^{-1}$ | 9.02 $\pm 8e^{-2}$ | 8.26 $\pm 8e^{-2}$ | 8.90 $\pm 2e^{-2}$ | **7.52** $\pm 2e^{-2}$ | 0.440 $\pm 3e^{-3}$ | 0.386 $\pm 6e^{-3}$ | 0.426 $\pm 7e^{-3}$ | 0.389 $\pm 3e^{-4}$ | **0.495** $\pm 5e^{-3}$ |

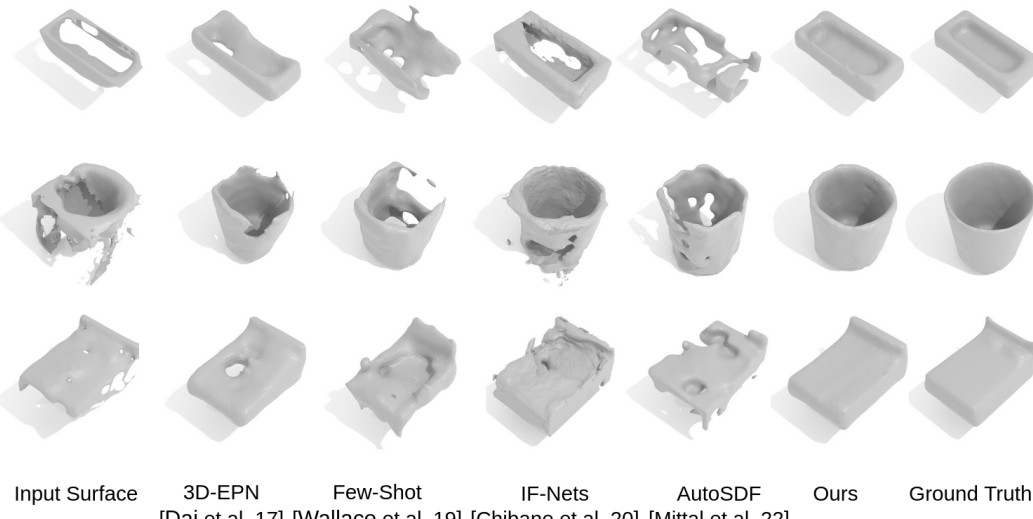

Input Surface    3D-EPN [Dai et al. 17]    Few-Shot [Wallace et al. 19]    IF-Nets [Chibane et al. 20]    AutoSDF [Mittal et al. 22]    Ours    Ground Truth

Figure 6: Shape completion on real-world ScanNet [10] object scans. Our method to learn mappings to multi-resolution learnable patch priors enables more coherent shape completion on novel categories.

Table 3: Ablation study on different patch resolutions. A multi-resolution approach gains benefits from both global and local reasoning.

| | ShapeNet [5] | | | | ScanNet [10] | | | |
|---|---|---|---|---|---|---|---|---|
| | Inst-CD↓ | Cat-CD↓ | Inst-IoU↑ | Cat-IoU↑ | Inst-CD↓ | Cat-CD↓ | Inst-IoU↑ | Cat-IoU↑ |
| Ours ($32^3$ priors only) | 11.97 $\pm 3e^{-2}$ | 11.62 $\pm 1e^{-2}$ | 0.35 $\pm 1e^{-3}$ | 0.37 $\pm 1e^{-3}$ | 10.40 $\pm 3e^{-2}$ | 11.33 $\pm 1e^{-1}$ | 0.41 $\pm 2e^{-3}$ | 0.39 $\pm 5e^{-3}$ |
| Ours ($8^3$ priors only) | 4.89 $\pm 3e^{-2}$ | 4.92 $\pm 3e^{-2}$ | 0.61 $\pm 4e^{-4}$ | 0.62 $\pm 1e^{-3}$ | 7.67 $\pm 3e^{-2}$ | 7.84 $\pm 3e^{-2}$ | 0.49 $\pm 6e^{-3}$ | 0.49 $\pm 2e^{-3}$ |
| Ours ($4^3$ priors only) | 4.45 $\pm 1e^{-2}$ | 4.50 $\pm 1e^{-2}$ | **0.64** $\pm 4e^{-3}$ | 0.64 $\pm 2e^{-2}$ | **7.37** $\pm 3e^{-2}$ | 7.63 $\pm 5e^{-2}$ | 0.48 $\pm 4e^{-3}$ | 0.48 $\pm 7e^{-3}$ |
| Ours | **4.23** $\pm 4e^{-2}$ | **4.27** $\pm 5e^{-2}$ | **0.64** $\pm 1e^{-3}$ | **0.65** $\pm 1e^{-3}$ | 7.38 $\pm 6e^{-2}$ | **7.52** $\pm 2e^{-2}$ | **0.50** $\pm 9e^{-3}$ | **0.50** $\pm 5e^{-3}$ |

## 4.4 Ablation Analysis

**Does multi-resolution patch learning help shape completion for novel categories?** In Table 3, we evaluate shape completion with each resolution in comparison with our multi-resolution approach. Learning only global shape priors (i.e., $32^3$) tends to overfit to seen train categories, while the local patch resolutions can provide more generalizable priors. Combining all results in complementary feature learning for the most effective shape completion results.

**Does cross-attention to learn local priors help?** We evaluate our approach to learn both local priors and their correlation to input observations with cross-attention in Table 4, which shows that this enables more effective shape completion on unseen categories. Here, the *no attention* experiment replaces the attention score calculation in Eq. 1 by using MLPs (on concatenated input/prior features) to predict weights for each input-prior pair.

Table 4: Ablation study on attention used to learn input and patch prior correlations.

|  | ShapeNet [5] | | | | ScanNet [10] | | | |
|---|---|---|---|---|---|---|---|---|
|  | Inst-CD↓ | Cat-CD↓ | Inst-IoU↑ | Cat-IoU↑ | Inst-CD↓ | Cat-CD↓ | Inst-IoU↑ | Cat-IoU↑ |
| Ours (no attention) | 4.90 | 4.98 | **0.61** | 0.62 | 7.80 | 8.09 | **0.49** | 0.48 |
| Ours | **4.69** | **4.74** | **0.61** | **0.63** | **7.58** | **7.84** | 0.48 | **0.49** |

Table 5: Effect of synthetic pre-training on real-world ScanNet [10] object completion vs. training from scratch.

|  | Inst-CD↓ | Cat-CD↓ | Inst-IoU↑ | Cat-IoU↑ |
|---|---|---|---|---|
| Scratch | 7.61 | 7.73 | **0.50** | **0.50** |
| Ours | **7.38** | **7.52** | **0.50** | **0.50** |

**What is the effect of synthetic pre-training for real scan completion?** Table 5 shows the effect of synthetic pre-training for shape completion on real scanned objects. This encourages learning more robust priors to output cleaner local structures as given in the synthetic data, resulting in improved performance on real scanned objects.

**Does learning the priors help completion?** In Table 6, we evaluate our learnable priors in comparison with using fixed priors (by mean-shift clustering of train objects) for shape completion on ShapeNet [5]. Learned priors receive gradient information to adapt to best reconstruct the shapes, enabling improved performance over a fixed set of priors.

Table 6: Ablation on learnable priors in comparison with fixed priors on ShapeNet [5].

|  | Inst-CD↓ | Cat-CD↓ | Inst-IoU↑ | Cat-IoU↑ |
|---|---|---|---|---|
| Ours (fixed priors) | 4.31 | 4.34 | **0.64** | **0.65** |
| Ours | **4.23** | **4.27** | **0.64** | **0.65** |

**What is the effect of the train category split?** Our novel category split was designed based on the number of objects in each category, to mimic real-world scenarios where object categories with larger numbers of observations are used for training.

To show that our approach is independent of the category splitting strategy, we add two new settings for evaluation on ShapeNet [5]. We use the same overall categories (26) as originally, then randomly shuffle them for train/novel categories. Table 7 shows that all the splits achieve similar results, which indicates that our approach is robust across different splits. It is because the learned patch priors share universal partial informa-

Table 7: Ablation study on train category split on ShapeNet [5].

|  | Inst-CD↓ | Cat-CD↓ | Inst-IoU↑ | Cat-IoU↑ |
|---|---|---|---|---|
| Category Split 1 | 4.30 | 4.29 | 0.65 | 0.66 |
| Category Split 2 | 4.19 | 4.25 | 0.68 | 0.67 |
| Ours | 4.23 | 4.27 | 0.64 | 0.65 |

tion among categories, which reduces the dependency on train categories for shape completion tasks. Figure 7 shows qualitative results for novel category samples in the alternative splits.

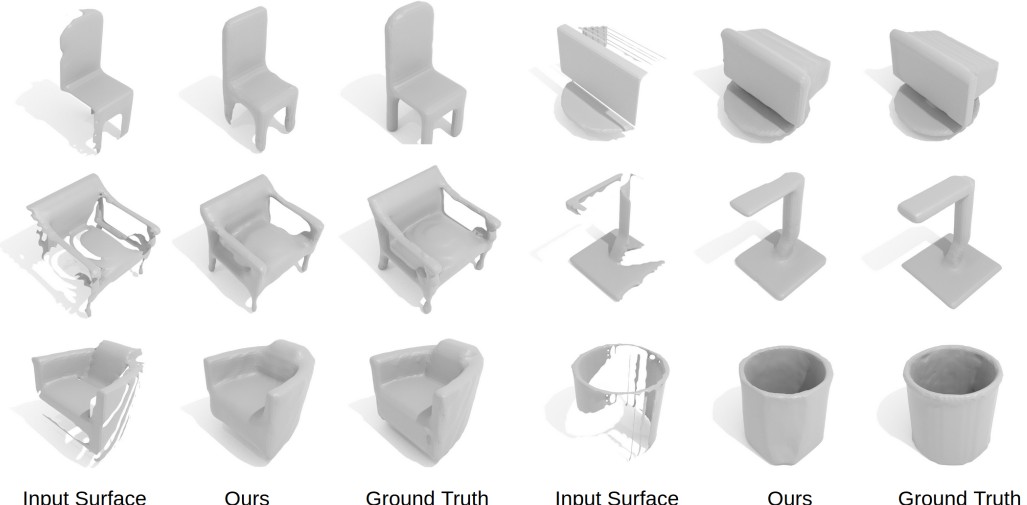

| Input Surface | Ours | Ground Truth | Input Surface | Ours | Ground Truth |

Figure 7: Qualitative results for shape completion on ShapeNet [5] for novel category samples in alternative train category splits.

## 4.5 Limitations

While PatchComplete has presented a promising step towards learning more generalizable shape priors, various limitations remain. For instance, output shape completion is limited by the dense voxel grid resolution in representing fine-scale geometric details. Additionally, detected object bounding boxes are required for real scan data as input to independent shape completion predictions, while formulation considering other objects in the scene or an end-to-end framework could learn more effectively from global scene contextual information.

## 4.6 Societal Impact

This work proposes a method for 3D completion for unseen categories with less dependency on data capturing, with potential application to robotics or content creation scenarios. For instance, a robot could potentially more efficiently grasp unseen objects when estimating complete shape structure; or self-driving cars could perform obstacle avoidance by robust 3D shape estimation for a variety of objects. While potential real-world applications are promising, this requires careful consideration of personal data privacy, as well as any biases embedded in the training dataset.

## 5 Conclusion

We have proposed PatchComplete to learn effective local shape priors for shape completion that enables robust reconstruction for novel class categories at test time. This enables learning shared local substructures across a variety of shapes that can be mapped to local incomplete observations by cross attention, with multi-resolution fusion producing coherent geometry at global and local scales. This enables robust shape completion even for unseen categories with different global structures, across synthetic as well as challenging real-world scanned objects with noise and clutter. We believe that such robust reconstruction of real scanned objects takes an important step towards understanding 3D shapes, and hope this inspires future work in understanding real-world shapes and scene structures.

## Acknowledgements

This project is funded by the Bavarian State Ministry of Science and the Arts and coordinated by the Bavarian Research Institute for Digital Transformation (bidt).

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
