# OpenReview forum: "PatchComplete: Learning Multi-Resolution Patch Priors for 3D Shape Completion on Unseen Categories"
_NeurIPS.cc/2022/Conference — NeurIPS 2022 Accept_

### Official Review · Reviewer_oDR1 · 2022-06-17

**Rating:** 6
**Confidence:** 3
**Soundness:** 3 good
**Presentation:** 4 excellent
**Contribution:** 2 fair

**Summary:**

The submission introduces PatchComplete for 3D object-level shape completion from a partial input SDF voxel grid, with a focus on generalizing to unseen object categories (not just unseen instances). It evaluates on the synthetic ShapeNet and the real-world ScanNet, which contains 3D reconstructions of rooms (ScanNet annotations are used to segment out objects; pseudo-GT full object shapes are obtained via Scan2CAD). The paper contributes a scheme to learn patch-level shape priors. It also contributes a method that associates an input incomplete shape with that patch-level prior and then reconstructs the complete shape as a (learned) merging of that associated/weighted prior information. Comparisons are done to a diverse set of prior work and PatchComplete outperforms them, especially on unseen categories. A number of ablations are also included.

**Questions:**

Questions for the rebuttal:

1) It is not clear to me whether the accompanying code will be released.

2) I have several questions about Table 3:
a) The ablation of which resolutions to use in the multi-resolution scheme (Table 3) shows rather marginal improvements when going from only the finest resolution (4^3) to all three resolutions (Ours; 4^3, 8^3, 32^3). Are really both 8^3 AND 32^3 necessary? How does 4^3+8^3 and 4^3+32^3 perform?
b) What are the standard deviations for these numbers? Are the marginal improvements significant?
c) Especially on ScanNet, which is the more interesting dataset compared to ShapeNet for this task, shows results that are on par for 4^3 only and for Ours. I am not really convinced that this supports the claim that using multi-resolution leads to the "most effective shape completion results" (line 216).

3) An ablation of the concatenation in Eq. 3 could be added, to get a sense of how much the patch prior and how much the input encoding can achieve on their own, and what their respective issues are. I.e., use only Q^R_i in one ablation and use only the Attention() in another ablation. This is not crucial though.

4) What is the "no attention" ablation in Table 4? What is used instead of attention? Only Q^R_i directly, as I suggested in 3)? What else is changed?

5) An overall ablation that uses fixed priors, uses no pre-training, uses no attention, and only uses resolution 4^3 would be interesting. How much do these tweaks contribute to the overall performance?

Suggestions for improvement that are not relevant for the rebuttal:

- I suggest to add qualitative results for the ablations, at least in the supplement.

- How many shapes/priors does {T^c} contain? I.e. how many representative samples (in total) are there out of how many total shapes {G^c}?

- Some related works on deep implicit local geometry representations could be added:

* Genova et al. Local Deep Implicit Functions for 3D Shape, CVPR 2020
* Charbra et al. Deep Local Shapes: Learning Local SDF Priors for Detailed 3D Reconstruction, ECCV 2020
* Takikawa et al. Neural Geometric Level of Detail: Real-Time Rendering With Implicit 3D Shapes, CVPR 2021
* Jiang et al. Local Implicit Grid Representations for 3D Scenes, CVPR 2020
* Deng et al. NASA: Neural Articulated Shape Approximation, ECCV 2020

- There's a superfluous period in line 95.

**Limitations:**

- Limitations are mentioned.

- Societal impact is discussed.

**Strengths And Weaknesses:**

_Positives_

The method is novel and enables good generalization to unseen categories, which is particularly useful for static 3D reconstruction of unseen environments with systems like KinectFusion.

The paper is written well in terms of language and clarity.

The experimental evaluation is done well. The ablations consider a multitude of factors of the method.

_Negatives_

I don't see much on the technical level that could potentially be used in other areas of CV/ML. The method consists of a core (attention into a learned patch prior) and a number of tweaks, which are evaluated in the ablations. The core part is a simple version of (non-sparse) dictionary learning.

---

> ### Author Response · Authors · 2022-08-02
> **Response to Reviewer oDR1.**
>
> Thank you for your valuable review; we are glad that our method was found to be 'novel' and to enable 'good generalization to unseen categories', with 'experimental evaluation [that] is done well'.
>
> **Applications.** Our method focuses on the problem of shape completion on objects from unseen categories.
> We believe our approach of disentangling the shape reconstruction task to learning local substructures has the potential to be applied to various 3D reconstruction tasks for unseen objects or environments in the future, for instance, single-view shape reconstruction, or 3D scene completion and reconstruction.
>
> **Code release.** We will publicly release the code and data.
>
> **Multi-resolution ablation.** We evaluate alternative multi-resolution combinations in Table 1, which shows that all resolutions benefit the more detailed chamfer evaluation (whereas IoU only penalizes non-intersections, rather than how far the predictions are from the GT object).
>
> &nbsp;&nbsp;&nbsp;&nbsp;Table 1: Ablation study of patch resolutions on synthetic ShapeNet data (CD × ${10}^{2}$).
> | | Inst-CD$\downarrow$ | Cat-CD$\downarrow$ | Inst-IoU$\uparrow$ | Cat-IoU$\uparrow$|
> |---|:---|:---|:---|:---|
> |Ours (${4}^{3}$ with ${32}^{3}$)  | 4.30 | 4.35 | 0.642 | 0.651|
> |Ours (${4}^{3}$ with ${8}^{3}$) | 4.35 | 4.42 | **0.644** | **0.654**|
> |Ours (all resolutions) | **4.23** | **4.27** | **0.644** | **0.654**|
>
> We additionally evaluate the standard deviations of the multi-resolution ablations in Table 2 (note that the single-resolution evaluations were only run once for the main paper, and so the averages have changed the values slightly). The multi-resolution improvements are significant and consistent in both settings.
>
> &nbsp;&nbsp;&nbsp;&nbsp;Table 2: Ablation study on patch resolution (inc. standard deviation, CD × ${10}^{2}$).
>
> ||ShapeNet||||ScanNet||||
> |---|:---|:---|:---|:---|:---|:---|:---|:---|
> | | Inst-CD$\downarrow$ | Cat-CD$\downarrow$ | Inst-IoU$\uparrow$ | Cat-IoU$\uparrow$|Inst-CD$\downarrow$ | Cat-CD$\downarrow$ | Inst-IoU$\uparrow$ | Cat-IoU$\uparrow$|
> |Ours (${32}^{3}$ priors only) | 11.97 $\pm3{e}^{-2}$ |11.62 $\pm1{e}^{-2}$ | 0.35 $\pm1{e}^{-3}$|0.37 $\pm1{e}^{-3}$|10.40 $\pm3{e}^{-2}$|11.33 $\pm1{e}^{-1}$|0.41 $\pm2{e}^{-3}$|0.39 $\pm5{e}^{-3}$|
> |Ours (${8}^{3}$ priors only) | 4.89  $\pm3{e}^{-2}$ | 4.92 $\pm3{e}^{-2}$ | 0.61 $\pm4{e}^{-4}$ | 0.62 $\pm1{e}^{-3}$|7.67  $\pm3{e}^{-2}$|7.84 $\pm3{e}^{-2}$|0.49 $\pm6{e}^{-3}$|0.49 $\pm2{e}^{-3}$|
> |Ours (${4}^{3}$ priors only) | 4.45 $\pm1{e}^{-2}$ | 4.50 $\pm1{e}^{-2}$ | **0.64** $\pm4{e}^{-3}$ | 0.64 $\pm2{e}^{-2}$ | **7.37** $\pm3{e}^{-2}$|7.63 $\pm5{e}^{-2}$ | 0.48 $\pm4{e}^{-3}$ | 0.48 $\pm7{e}^{-3}$|
> |Ours|**4.23** $\pm4{e}^{-2}$|**4.27**$\pm5{e}^{-2}$|**0.64**$\pm1{e}^{-3}$|**0.65** $\pm1{e}^{-3}$|**7.37** $\pm7{e}^{-2}$|**7.49** $\pm4{e}^{-2}$|**0.50**$\pm9{e}^{-3}$|**0.50**$\pm5{e}^{-3}$|
>
> **Eq. 3 Ablation on concatenation.** We evaluate the effectiveness of concatenation in *Eq.3* in Table 3, considering the attention-based term only (the core of our approach). We note that when excluding the attention-based term, this does not consider local patches anymore and becomes similar to the encoder-decoder training of 3D-EPN.
> As the attention-based learning of correspondence to local priors is the core of our approach, this produces the most relative benefit, with a slight improvement when combining the terms together.
>
> &nbsp;&nbsp;&nbsp;&nbsp;Table 3: Concatenation ablation study for each term in Eq.3 on the ShapeNet dataset (CD × ${10}^{2}$).
> | | Inst-CD$\downarrow$ | Cat-CD$\downarrow$ | Inst-IoU$\uparrow$ | Cat-IoU$\uparrow$|
> |---|:---|:---|:---|:---|
> |3D-EPN | 5.48 | 5.58 | 0.582 | 0.594|
> |Ours (attention term only) | 4.25 | 4.29 | 0.640 | 0.650|
> |Ours | **4.23** | **4.27** | **0.644** | **0.654**|
>
> **'No attention' in Table 4.** In Table 4, the 'no attention' experiment replaces the attention score calculation in Eq. 1 by using MLPs (on concatenated input/prior features) to predict weights for each input-prior pair.
>
> **Ablation for fixed priors, no pre-training, and no attention on ${4}^{3}$ priors only.** We evaluate this scenario in Table 4, which produces significantly worse results due to the lack of learnable priors in combination with attention.
>
> &nbsp;&nbsp;&nbsp;&nbsp;Table 4: Evaluation for fixed priors, no pre-training, and no-attention on ${4}^{3}$ priors only (CD × ${10}^{2}$).
> | | Inst-CD$\downarrow$ | Cat-CD$\downarrow$ | Inst-IoU$\uparrow$ | Cat-IoU$\uparrow$|
> |---|:---|:---|:---|:---|
> |Ours (fixed priors, no pre-training, and no-attention on ${4}^{3}$ priors only) | 9.53 | 9.73 | 0.35 | 0.37 |
> |Ours | **7.37** | **7.49** | **0.50** | **0.50** |
>
> **Misc.** Thanks for the suggestions on visualization, references, and formation, we have modified our paper for the final version. We use 112 shape priors in our method, which are clustered from the 3202 train shapes, and will include more clarification in the final paper.

---

### Official Review · Reviewer_z3XK · 2022-07-11

**Rating:** 6
**Confidence:** 4
**Soundness:** 3 good
**Presentation:** 3 good
**Contribution:** 3 good

**Summary:**

The paper tackles a challenging problem - the performance degradation on unseen categories in shape completion tasks. Inspired by the observation that different categories may share similar local geometric structures, the authors propose a method to learn multi-resolution patch priors and then reconstruct the complete shape using the recursively fused features. The results show a significant performance improvement in novel-category shape completion compared to baseline methods, and ablations have verified the importance of each component.

**Questions:**

1. I wonder about the criteria for splitting the datasets. In my opinion, the partition of the data has an enormous impact on performance (e.g., it is hard to complete a chair if we only have sofas as priors). If you randomly choose the training categories, I think it would be better to use cross-validation to show the impact of how the data is divided or the generalization performance.

2. Most of the categories in the paper and supplementary only have simple geometry, such as bed or bathtub. Although you mention the lack of fine-scale geometric details in the results, I am still curious about performance on more complex data (such as the chair or bookshelf you mentioned in the video).

3. What is the impact of the number of priors? In the ablation part, the authors show the results on different patch resolutions, and it seems only 4^3 priors can achieve competitive performance. Can I assume that local path priors play a major role, and it may have less diversity? So it may have redundancy, and the boundary of the number of priors is meaningful in real-world applications. Moreover, it would be better to show the memory usage for storing the priors.

4. How about the domain independence of the proposed method? Have you tried the cross-dataset experiments? More specifically, can you train priors on ShapeNet, and test using ScanNet? Logically speaking, there should be a reasonable result.
5. By the way, why choose TSDF as the input shape representation? Is it because it is easy to perform 3D convolution operations? Is it possible to change TSDF to point cloud?

**Limitations:**

Tthe paper bills itself that it can complete shapes for entirely unseen categories, but the authors do not illustrate the criteria of splitting the categories in the dataset, making the generalization of the proposed method on unseen categories obscure. It would be better to show more results on this to support the claim.

Another concern is the performance on categories with more complex geometries, logically, learning over the patch level could be helpful to improve the local geometric details in the synthesized results. Although the authors mention it in the limitations, the results of the method in its current form do not fully show the power of leveraging patch-level priors.

**Strengths And Weaknesses:**

Learning patch-level priors and the fusion pipeline of different resolution patch priors is moderately novel to me. The pipeline in this submission is technically sound. The submission is clearly written and well organized. The authors also provide the source code in supplementary material for better clarity of the techincal details. BTW, I’m a bit impressed that the completed shapes are mostly watertight and have continuous geometry.

In general, the evaluation part of the manuscript is good to me. The numerical results show a significant performance improvement over the baseline methods on both synthetic and real scan data. However, as the authors mentioned in limitations, visual and qualitative results on categories with fine geometric structures are not shown, which could have been a strong support to the merit of the proposed method. For the ablation study, the authors answer the importance of each designed component and show convincing numerical results of the effectiveness.

---

> ### Author Response · Authors · 2022-08-02
> **Response to Reviewer z3XK.**
>
> Thank you for your helpful feedback, and we are glad that our patch-level priors and multi-resolution fusion was found to be 'novel' and 'technically sound'.
>
> **Cross Validation.** Our novel category split was designed based on the number of objects in each category, to mimic real-world scenarios where object categories with larger numbers of observations are used for training.
> To show that our approach is independent of the category splitting strategy, we add two new settings for evaluation on ShapeNet.
> We use the same overall categories (26) as originally, then randomly shuffle them for train/novel categories.
> To consider chair/bookshelf performance, categories are shuffled until these classes appear in the test split.
> We then consider two more splits where chair and bookshelf appear in the novel category split, respectively.
> Table 1 shows that our approach is robust across these splits.
>
> For Split 1, the 8 novel testing categories are *trash bin, bed, piano bench, **chair**, monitor, lamp, laptop, washing machine*.
> For Split 2, the 8 novel testing categories are *basket, **bookshelf**, bowl, cabinet, laptop, pot, sofa, stove*.
>
> &nbsp;&nbsp;&nbsp;&nbsp;&nbsp;&nbsp;&nbsp;&nbsp;Table 1: Category split ablation on ShapeNet (CD × ${10}^{2}$).
> | | Inst-CD$\downarrow$ | Cat-CD$\downarrow$ | Inst-IoU$\uparrow$ | Cat-IoU$\uparrow$|
> |:---|:---|:---|:---|:---|
> |Split 1 | 4.30 | 4.29 | 0.65 | 0.66|
> |Split 2 | 4.19 | 4.25 | 0.68 | 0.67|
> |Ours | 4.23 | 4.27 | 0.64 | 0.65|
>
> **Performance on Complex Data** From the cross-validation experiments, we see that our method can effectively handle the more complex geometry of chairs and bookshelves in Table 2. We will include additional qualitative results in the final version.
>
> &nbsp;&nbsp;&nbsp;&nbsp;&nbsp;&nbsp;&nbsp;&nbsp;Table 2: Quantitative results for chair from Split 1 and bookshelf from Split 2 (CD × ${10}^{2}$).
> | | CD$\downarrow$ | IoU$\uparrow$|
> |---|:---|:---|
> |Chair | 4.65 | 0.64|
> |Bookshelf | 4.15 | 0.61|
>
> **Impact of the Number of Priors.** We evaluate the effect of different numbers of priors on ShapeNet data in Table 3 (with 50\% priors and 150\% priors).
> We see that performance degrades with 50\% priors, while further increasing the prior number reaches a performance plateau (and requires additional storage). In our approach, our prior storage takes 14.68 MB in memory.
>
> &nbsp;&nbsp;&nbsp;&nbsp;&nbsp;&nbsp;&nbsp;&nbsp;Table 3: Ablation on the number of shape priors (CD × ${10}^{2}$).
> | | Inst-CD$\downarrow$ | Cat-CD$\downarrow$ | Inst-IoU$\uparrow$ | Cat-IoU$\uparrow$|
> |---|:---|:---|:---|:---|
> |Ours (50\% priors)  | 4.41 | 4.45 | 0.632 | 0.640|
> |Ours | 4.23 | **4.27** | **0.644** | **0.654**|
> |Ours (150\% priors)  | **4.22** | 4.30 | 0.638 | 0.647|
>
> **Domain Independence.** Training on ShapeNet and testing directly on ScanNet is a challenging task, as ShapeNet objects are in isolation, whereas ScanNet objects can contain background clutter around them.
> When doing so without any fine-tuning, our method can still provide reasonable results, and achieves performance on par with state-of-the-art methods that have been fine-tuned on ScanNet data, as shown in Table 4.
>
> &nbsp;&nbsp;&nbsp;&nbsp;&nbsp;&nbsp;&nbsp;&nbsp;Table 4: Ablation study on domain independence (CD × ${10}^{2}$).
> | | Inst-CD$\downarrow$ | Cat-CD$\downarrow$ | Inst-IoU$\uparrow$ | Cat-IoU$\uparrow$|
> |---|:---|:---|:---|:---|
> |Best-performing SOTA baseline | 8.12 | 8.26 | 0.44 |0.44|
> |Ours (w/o finetuning)  | 8.17 | 8.44 | 0.44 | 0.46|
> |Ours (w/ finetuning) | 7.37 | 7.49 | 0.50 | 0.50|
>
> **TSDF Representation.** We use a TSDF representation, which is common in 3D scanning and capture methods (e.g., Volumetric Fusion, KinectFusion, etc.), and TSDFs provide information in empty regions about the distance to object surfaces, along with the sign indicating known/unknown regions with respect to the camera.
> Additionally, the volumetric representation provides a natural spatial correlation between priors and inputs.
> Point cloud inputs could potentially be processed by converting to volumetric grids for testing, or extracting point-based local input feature regions for the input-prior association during training.

---

### Official Review · Reviewer_ngNb · 2022-07-16

**Rating:** 6
**Confidence:** 4
**Soundness:** 3 good
**Presentation:** 2 fair
**Contribution:** 3 good

**Summary:**

This paper proposes a deep network architecture for performing 3D shape completion on unseen shape categories by learning local geometric priors at multiple scales. By learning sets of features for distinct categories on patches sampled at different resolutions and fusing these features with those computed from an input partial scan from an unseen category, the network is able to leverage the local information  to achieve a better reconstruction. The method is validated qualitatively and quantitatively against state-of-the-art baselines, and a brief ablation study is provided to justify the multi-resolution aspect.

**Questions:**

Why is the softmax denominator in (1) $d/2$? This seems to depart from the usual $\sqrt d$.

How are the partial shapes inputted to the pipelines as SDFs? Aren't the partial shapes generically not closed surfaces?

Why do the ground truth synthetic surfaces in Figure 5 have artifacts and imperfections?

To what extent is the method robust to some of the hyperparameters used? For example, in particular how important is the choice of resolutions and the number of layers in the pyramid in the multi-scale approach? Similarly, how are the coefficients for the loss terms chosen?

How is the clustering to initialize the shape priors performed? Is it done directly on the SDF representations?

**Limitations:**

Limitations are sufficiently discussed.

**Strengths And Weaknesses:**

This paper makes a nice step towards training shape reconstruction models that can handle out-of-distribution examples from unseen categories. The multi-scale approach is sensible, and the experimental results validate its effect.

The results that are shown are all quite low frequency, with the shapes having fairy limited local variation. I wonder why this method is unable to handle more detailed geometries. Is this due to the fact that such textures are not seen in the training categories, or is there a different bottleneck?

The method description is pretty heavy on notation and somewhat difficult to follow. It would be helpful to clearly define the different objects that are considered and clearly distinguish between geometry representations and latent high-dimensional encodings. To this end it's also important to fix subtle typos in the notation, e.g. $S$ should be $\mathcal S$ on L107.

---

> ### Author Response · Authors · 2022-08-02
> **Response to Reviewer ngNb.**
>
> Thank you for your constructive review, and we are glad that our multi-resolution approach was found to be 'sensible' with 'experimental results [that] validate its effect'.
>
> **Detailed Geometry.** In order to further measure the potential for detail, we have conducted two cross-validation experiments. Rather than considering the largest-represented categories for training, as in our paper setup, we arbitrarily shuffle categories to obtain a setup where chairs and bookshelves lie in the novel category test set (following z3XK's suggestion). Here, we see that our approach can maintain effective geometric representation for these complex categories, with 0.64 IoU and 4.65 x ${10}^{-2}$ CD for chairs, and 0.61 IoU and 4.15 x ${10}^{-2}$ CD for bookshelves.
>
> **Softmax Denominator.** Empirically, we found  $d/2$ rather than $\sqrt{d}$ to provide very slightly better performance.
>
> **Input Partial Shapes.** We use an SDF representation following that of popular volumetric 3D reconstruction and scanning methods (e.g., Volumetric Fusion [1], KinectFusion [2], BundleFusion [3]), and leveraged by alternative shape and scene completion methods (3D ShapeNets [4], 3D-EPN [5], SSCNet [6]). Here, input depth frames are fused into an SDF grid where the sign denotes in-front-of a surface (known empty) vs. behind a surface (unknown), rather than inside-outside.
>
> **Fig 5. GT.** We generate ground truth SDFs for ShapeNet by applying virtual rendering and fusion on synthetic shape meshes, following Occupancy Networks [7]. The volumetric resolution can lead to small discretization artifacts and may not be able to capture very fine-scale details.
>
> **Influence of Hyperparameters.** We considered patch resolutions of ${4}^{3}$, ${8}^{3}$, ${16}^{3}$, and ${32}^{3}$. We found ${16}^{3}$ and ${8}^{3}$ to perform very similarly (variance of $8{e}^{-6}$ IoU and $6{e}^{-5}$ CD), and used ${8}^{3}$ to potentially resolve more detailed patches.
>
> For the multi-resolution pyramid, we consider different combinations with  ${4}^{3}$ (which provided the best single-resolution results) in Table 1. Here, our multi-resolution approach performs the best with a combination of global and local reasoning.
>
> The performance variation between loss coefficients tested produced a variance of $2{e}^{-4}$ in IoU and $2{e}^{-5}$ in CD; we used the coefficients that produced the best validation results.
>
> &nbsp;&nbsp;&nbsp;&nbsp;&nbsp;&nbsp;&nbsp;&nbsp;Table 1: Ablation study of patch resolutions on synthetic ShapeNet data (CD × ${10}^{2}$).
> | | Inst-CD$\downarrow$ | Cat-CD$\downarrow$ | Inst-IoU$\uparrow$ | Cat-IoU$\uparrow$|
> |---|:---|:---|:---|:---|
> |Ours (${4}^{3}$ with ${32}^{3}$)  | 4.30 | 4.35 | 0.642 | 0.651|
> |Ours (${4}^{3}$ with ${8}^{3}$) | 4.35 | 4.42 | **0.644** | **0.654**|
> |Ours (all resolutions) | **4.23** | **4.27** | **0.644** | **0.654**|
>
> **Prior Clustering.** We cluster the TSDF representations of the train shapes (a truncation of 2.5 voxels) by mean shift clustering to generate the shape priors.
>
> **Notation.** Thank you for your suggestions, we have made a pass through the paper to clarify the notation.
>
> **Reference**
>
> [1] Volumetric method for building complex models from range images. [Curless and Levoy 96]
>
> [2] Kinectfusion: Real-time dense surface mapping and tracking. [Newcombe et al. 11]
>
> [3] Bundlefusion: Real-time globally consistent 3d reconstruction using on-the-fly surface reintegration. [Dai et al. 17]
>
> [4] 3d shapenets: A deep representation for volumetric shapes. [Wu et al. 15]
>
> [5] Shape completion using 3d-encoder-predictor cnns and shape synthesis. [Dai et al. 17]
>
> [6] Semantic scene completion from a single depth image. [Song et al. 17]
>
> [7] Convolutional occupancy networks. [Peng et al. 20]

---

### Official Review · Reviewer_Bd3i · 2022-07-25

**Rating:** 6
**Confidence:** 3
**Soundness:** 3 good
**Presentation:** 3 good
**Contribution:** 2 fair

**Summary:**

This paper proposed a 3D Shape reconstruction method using local patch priors. The proposed method uses the multi-resolution patch priors based on the observation that within a 3D structure, some details are repetitive and often easier to be constructed first. The paper is well written and balanced

**Questions:**

* the method is indeed effective over the test case, how about the efficiency? what is the average runtime compared to other signal-pass and one-resolution methods?

**Limitations:**

Natural limitations as the author also pointed out, the bounding boxes are required for the scan, which means this method is constrained to a predefined space and is not directly translatable to use in the wild.

**Strengths And Weaknesses:**

+ the presentation of the paper is well put together

+ the illustration of the paper is detailed and easy to understand

+ the idea of using local prior and multi-resolution is not necessarily brand new, but it is effective for the method the paper is proposing.

+ based on the paper's experiment results, it indeed improved some real-world 3D reconstruction

- it would be great if the author could showcase some reconstruction from the real world without a ground-truth scan. Just a static RGB photo and the 3D reconstructed object would help better demonstrate the strength of the proposed method.

---

> ### Author Response · Authors · 2022-08-02
> **Response to Reviewer Bd3i.**
>
> Thank you for your valuable review; we are glad that our method and presentation were found to be 'effective' and 'well put together'.
>
> **Time Efficiency.** We evaluate runtime efficiency in Table 1. Times are measured for each method for a single shape prediction (running with batch size of 1), averaged over 20 samples. Here, *Ours (${M}^{3}$ priors only)* denotes our approach with only single-resolution $M^3$ priors.
>
> &nbsp;&nbsp;&nbsp;&nbsp;&nbsp;&nbsp;&nbsp;&nbsp;&nbsp;Table 1: Quantitative comparison for testing time efficiency (s).
> |3D-EPN|Few-Shot|IF-Nets|AutoSDF|${4}^{3}$ priors only|${8}^{3}$ priors only|${32}^{3}$ priors only|Ours|
> |:---|:---|:---|:---|:---|:---|:---|:---|
> |0.015 | 0.004 | 0.421 | 0.958 | 0.025 | 0.017 | 0.016 | 0.063||
>
>
> **Additional Real-world Scenarios.** We are happy to show additional real-world results, in addition to the challenging real-world scenario or ScanNet scanned objects.

---

### Meta-Review · Area_Chair_dKtV · 2022-08-25

**Recommendation:** Accept
**Confidence:** Certain

**Metareview:**

This is an interesting paper on class-independent 3d shape completion. Reviewers agree that the paper has good quality and is moderately original. There were initially some questions about the level of generalization to new classes, but after a strong rebuttal all reviewers find the results compelling and all of them suggest acceptance. I agree with their assessment.

**Award:**

No

---

### Decision · Program_Chairs · 2022-09-14

Accept